# Recombinase Polymerase Amplification Combined with Real-Time Fluorescent Probe for *Mycoplasma pneumoniae* Detection

**DOI:** 10.3390/jcm11071780

**Published:** 2022-03-23

**Authors:** Tingting Jiang, Yacui Wang, Weiwei Jiao, Yiqin Song, Qing Zhao, Tianyi Wang, Jing Bi, Adong Shen

**Affiliations:** 1Baoding Key Laboratory for Precision Diagnosis and Treatment of Infectious Diseases in Children, Baoding Children’s Hospital, Baoding 071051, China; hbbdtt369@126.com (T.J.); jiaoweiwei@ccmu.edu.cn (W.J.); misssong20220322@163.com (Y.S.); 13933893379@163.com (Q.Z.); tansy1203@163.com (T.W.); 2Key Laboratory of Major Diseases in Children, Ministry of Education, National Key Discipline of Pediatrics (Capital Medical University), National Clinical Research Center for Respiratory Diseases, Beijing Key Laboratory of Pediatric Respiratory Infection Disease, Beijing Pediatric Research Institute, Beijing Children’s Hospital, Capital Medical University, National Center for Children’s Health, Beijing 100045, China; wangyacui123@mail.ccmu.edu.cn

**Keywords:** *Mycoplasma pneumoniae*, recombinase polymerase amplification, RPA, child, diagnosis

## Abstract

*Mycoplasma pneumoniae* (*M. pneumoniae*) is one of the major causes of community-acquired pneumonia, accounting for 20–40% of total cases. Rapid and accurate detection of *M. pneumoniae* is crucial for the diagnosis and rational selection of antibiotics. In this study, we set up a real-time recombinase polymerase amplification (RPA) assay to detect the conserved gene CARDS of *M. pneumoniae*. The amplification can be finished in 20 min at a wide temperature range from 37–41 °C. The limit of detection of RPA assay was 10 fg per microliter. Cross-reaction with commonly detected respiratory pathogens was not observed using RPA assay. Among clinical sputum samples, the detection rate of RPA assay and real-time PCR assay was 48.4% (92/190) and 46.3% (88/190), respectively (*p* = 0.68). Therefore, the RPA assay for *M. pneumoniae* detection is rapid and easy to use and may serve as a promising test for early diagnosis of *M. pneumoniae* infection.

## 1. Introduction

*Mycoplasma pneumoniae* (*M. pneumoniae*) is widely recognized as one of the major causes of respiratory infections in school-age children, accounting for 20–40% of community-acquired pneumonia (CAP) cases [1,2,3]. Although *M. pneumoniae* infection is generally a self-limiting disease that leads to mild and subclinical manifestations, sometimes it can cause severe pulmonary diseases and extra-pulmonary complications [4,5,6]. Recently, reports of life-threatening diseases associated with *M. pneumoniae* infection have been increasing in China [1,7]. Thus, rapid and simple diagnostic methods for *M. pneumoniae* are crucial to guide the appropriate treatment for the patients and reduce the proportions of severe cases.

Culture, serological and polymerase chain reaction (PCR) are three conventional methods for *M. pneumoniae* detection [8]. As the gold standard for the confirmation of *M. pneumoniae*, culture-based methods provide reliable evidence for infection. However, isolation culture is relatively insensitive, labor-intensive, cost-expensive, and usually takes 2–4 weeks to obtain results due to its fastidious and slow growth characteristics, limiting its application in clinical practice [8]. Serological technology is commonly used in routine work for its convenience. Nevertheless, the sensitivity and specificity of serological methods are susceptible to many confounding factors such as immunity status, children’s age, and testing kits [9]. In addition, reliable serological testing results depend on paired serum specimens at an interval of 1–2 weeks, which is not practical in clinics.

PCR-based technology exhibits excellent sensitivity and specificity as a fast diagnostic method [10,11]. In recent years, isothermal amplification-based methods, such as loop-mediated isothermal amplification (LAMP), multiple cross displacement amplification (MCDA), strand displacement amplification (SDA) and recombinase polymerase amplification (RPA), have been proved an alternative to routine PCR-based methods, in part because they are independent on sophisticated instruments. Without complicated thermal cycling, LAMP and MCDA assays can yield a positive result in 40–60 min [12,13,14]. However, these methods require 4–10 primers, and the primer design is typically complex, which relies on specialized personnel; hence, limiting its implementation in resource-limited settings. Recombinase polymerase amplification (RPA), a recombinase-based isothermal amplification, has been widely used for the detection of multiple pathogens due to its convenience and simplicity [15,16,17,18,19,20,21]. Distinct from LAMP and MCDA assays, only a pair of primers is needed to recognize the target sequence. Under the coordination of recombinase, polymerase, and single-strand binding proteins (SSB), RPA-based isothermal amplification can be carried out within 20 min at a temperature range of 37–42 °C [12].

This study aimed to establish an RPA assay for the rapid detection of *M. pneumoniae* and evaluate its applicability in clinical specimens.

## 2. Materials and Methods

### 2.1. Experiment Design

The primers and probe for RPA assay were designed based on the *M. pneumoniae* conserved CARDS gene sequences. The details of primers and probe used in this study were shown in Table 1. All the primers and probe were synthesized by TianyiHuiyuan Biotech Co. Ltd. (Beijing, China). RPA amplification kits were obtained from Amp-Future Co. Ltd. (Weifang, China). The genomic DNA of bacterial strains and clinical sputum samples were extracted using QIAamp DNA Mini Kit (Qiagen, Germany) and Universal DNA Extraction kit (Mole, China), respectively, according to the manufacturer’s instructions.

### 2.2. RPA Assay

The RPA technique was conducted in 50 μL reaction mixtures containing 29.5 μL buffer A, 2 μL each primer (10 μM), 0.6 μL probe (10 μM), 2.5 μL buffer B, 1 μL DNA templates extracted from isolated pathogens (5 μL for clinical specimens) and added distilled water (DW) to the total volume of 50 μL. Fluorescence acquisition of FAM (Carboxyfluorescein) fluorophore was performed every 30 s (end point reading) for 20 min at the temperature of 39 °C. For each run, reference strain M129 and DW were used as the positive control and negative control, respectively.

To determine the optimal reaction temperature of the RPA assay, we conducted the reaction with temperatures ranging from 37 °C to 41 °C for 20 min, and the amplicons were monitored by a fluorescence detector (Agilent Technology, AriaMx Real-time PCR, Santa Clara, CA, USA).

### 2.3. Specificity and Sensitivity of RPA Assay

The specificity of RPA assay was validated by testing genomic DNA (at least 1 ng/μL) from a panel of pathogens, including *M. pneumoniae* standard reference strains (M129), 14 commonly detected respiratory pathogens, and 5 other *Mycoplasma* species (Table 2).

The DNA template of reference strain M129 was quantified using Nanodrop ND-1000 (Thermo, Waltham, MA, USA), presented by A260/280 ratio. A ration ranging from 1.8 to 2.0 was acceptable. Then the DNA template was serially diluted to 100 pg, 10 pg, 1 pg, 100 fg, 10 fg, and 1 fg per microliter using DW. A volume of 1 μL was added into the RPA reaction. Five independent reactions were conducted for each dilution. The limit of detection (LOD) of RPA was determined as the last positive dilution.

### 2.4. The Application of RPA Assay in Clinical Specimens

To further evaluate the usefulness of RPA assay in clinical practice, sputum specimens from suspected *M. pneumoniae* infection patients were examined by RPA assay in parallel with commercial real-time PCR (Mole Bioscience, Jiangsu, China). The study was conducted according to the guidelines of the Declaration of Helsinki and approved by the Institutional Review Board (No. 2020-k-163). The written informed consent was waived, as the specimens used in this study were leftover samples from the clinical laboratory.

From August to December in 2021, patients hospitalized for suspected *M. pneumoniae* infection at Baoding Children’s Hospital were enrolled for detection. Sputum samples taken during hospitalization were used for commercial real-time PCR and RPA assay. SPSS version 23.0 (IBM) was used for all of the statistical analysis. Categorical variables were presented as frequencies. The diagnostic performance of RPA assay was presented as sensitivity and specificity. The sensitivity of the 2 methods was compared by the Chi-square test; *p*
*<* 0.05 was considered statistically significant. The kappa value of the real-time PCR and RPA assays was calculated. The Kappa value was interpreted as follows: poor, <0.4; moderate, 0.4–0.75; perfect, >0.75.

## 3. Results

### 3.1. Amplification Temperature Optimization

To determine the optimal reaction temperature, the RPA assay was conducted with 10 pg/μL DNA templates of M129 at a range of temperatures from 37–41 °C. As shown in Figure 1, RPA assays performed well with consistent amplification products in the 37–41 °C temperature range. Thus, the following RPA reaction in this study was conducted at a temperature of 39 °C.

### 3.2. Analytical Specificity and Sensitivity of RPA Assay

The specificity of the developed RPA assay was determined using respiratory-related pathogens. Fluorescence signals representing specific amplification of *M. pneumoniae* were obtained, and no cross-reactivity with other pathogens was found (Figure 2).

The sensitivity of the RPA assay was analyzed by detecting a series of 10-fold diluted DNA of reference strain M129 in 5 parallels. As shown in Figure 3, RPA assay demonstrated high sensitivity, with the LOD of 10 fg/uL. In particular, the same results were obtained for five independent reactions, illustrating the good reproducibility of the RPA assay.

### 3.3. Performance of RPA Assay in Clinical Specimens

The diagnostic performance of RPA assay was further validated in clinical sputum specimens, and the results were compared with that of commercial real-time PCR. Of the 190 specimens, the detection rate of RPA and commercial real-time PCR for *M. pneumoniae* was 48.4% (92/190) and 46.3% (88/190), respectively (χ^2^ = 0.17, *p* = 0.68). Furthermore, in comparison to real-time PCR, the sensitivity and specificity for RPA assay were 100% and 96.1%, respectively. The Kappa value of the 2 assays was 0.958 (Table 3).

## 4. Discussion

The increasing incidence of severe *M. pneumoniae* pneumonia poses an urgent requirement for rapid and reliable diagnostic approaches, especially in infrastructure-limited settings. The established RPA assay in this study is sensitive and specific which can be completed within 20 min at a wide temperature range (37 °C–41 °C) without the need for complicated equipment and procedures, which makes it a promising test for *M. pneumoniae* identification in clinical work.

The RPA assay developed here proved to have greater stringency for specificity, which allows for discrimination between *M. pneumoniae* and other common respiratory pathogens. In addition to specificity, the RPA assay also exhibited good sensitivity for *M. pneumoniae* detection. The LOD of the RPA assay was 10 fg per microliter. While the commercial real-time PCR used in this study is 500 copies/mL (about 45 fg per reaction) according to the manufacturer, suggesting that RPA assay is more sensitive. Compared with other isothermal techniques for detecting *M. pneumoniae*, such as MCDA (50 fg) and LAMP (600 fg) [13,14], the RPA assay established in this study also had better sensitivity. However, the better way to compare the sensitivity and specificity between two tests is using the same samples and the same experimental conditions. Further study is needed to evaluate the clinical performance of these different methods.

The prominent advantage of the RPA assay was its rapidity and simplicity, which contributes to the faster report of the results. It has been reported in previous publications that RPA technology can yield a positive result in 13 min when using DNA templates with concentrations of 100 copies/μL, while real-time PCR required approximately 56 min to get the result for the same specimens [22]. In this study, RPA assay could detect *M. pneumoniae* within 20 min, which is remarkably faster than real-time PCR (77 min) and other isothermal amplification methods, such as LAMP, SDA, MCDA (40–60 min) [12,13,14]. Additionally, an RPA-based assay does not have stringent requirements for amplification temperatures, which can be performed at a broad range of temperatures. Accumulating data of the RPA technique suggested that it performed well in a temperature range spanning from 22 °C to 45 °C, and no performance loss was observed [23,24]. In this study, we confirmed that RPA assay could be effectively conducted within the temperature range from 37 °C to 41 °C. Thus, an RPA assay does not need sophisticated equipment for precise temperature control, which makes it possible to carry out the test in resource-limited settings.

The readout of RPA amplification products can be achieved by lateral flow biosensor (LFB), agarose gel electrophoresis and real-time fluorescence detection. Owing to the additional purification process, agarose gel electrophoresis detection will lead to a loss of sensitivity. Li et al. reported that RPA-LFB allows amplification yields detection at concentrations of 100 fg DNA templates, which was 1000-fold more sensitive than that monitored by agarose gel electrophoresis [25]. LFB detector is more popular in the interpretation of RPA products for its rapid visualization, but aerosol pollution caused by open cover detection limits its wide application. Therefore, real-time fluorescence detection was selected for RPA yield reading in this study, which can ensure the sensitivity and obviate potential contamination generated by agarose and LFB during the process of opening tubes.

We further evaluated the diagnostic performance of the RPA assay in clinical sputum specimens. Our data revealed that the diagnostic performance of the established RPA assay was comparable to the commercial real-time PCR method, which is the most commonly used molecular detection method for various pathogens. The results were in line with preliminary reports [21,22,26,27], which showed that RPA assays performed well for different pathogens and can achieve concordance to real-time PCR techniques.

## 5. Conclusions

The RPA assay developed in this study is highly sensitive and specific for *M. pneumoniae* detection at a wide temperature range within 20 min. The characteristics of the RPA assay make it suitable for poorly equipped laboratories, which will serve as a promising tool for early detection of *M. pneumoniae*.

## Figures and Tables

**Figure 1 jcm-11-01780-f001:**
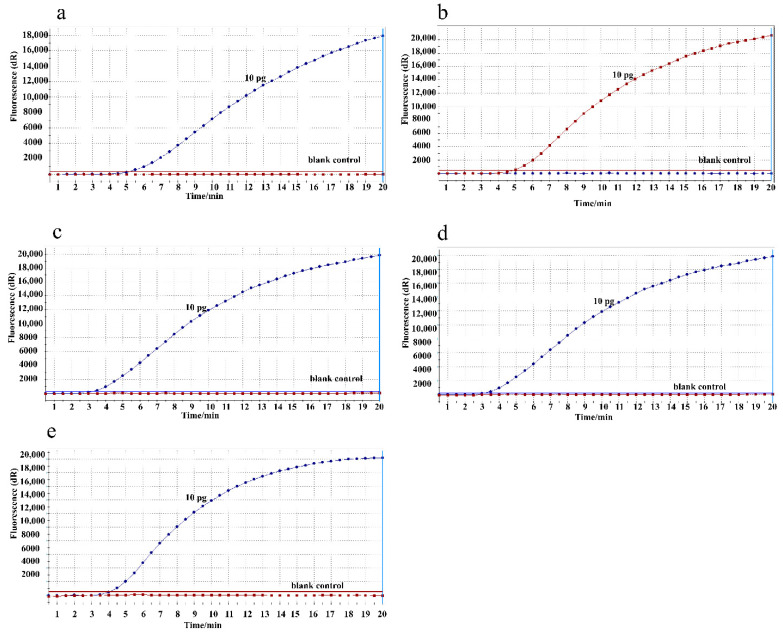
Temperature optimization for the RPA assay. a-e, RPA assay was conducted with 10 pg/μL M129 DNA templates in a broad range of amplification temperatures from 37 °C to 41 °C. (**a**) 37 °C; (**b**) 38 °C; (**c**) 39 °C; (**d**) 40 °C; (**e**) 41 °C.

**Figure 2 jcm-11-01780-f002:**
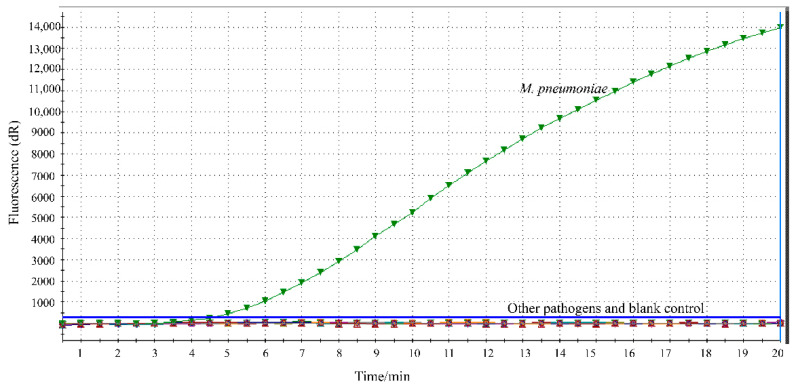
Analytical specificity of the RPA assay. The specific fluorescence signal was observed from *M. pneumoniae* and no signals were obtained from other pathogens (19) and blank control.

**Figure 3 jcm-11-01780-f003:**
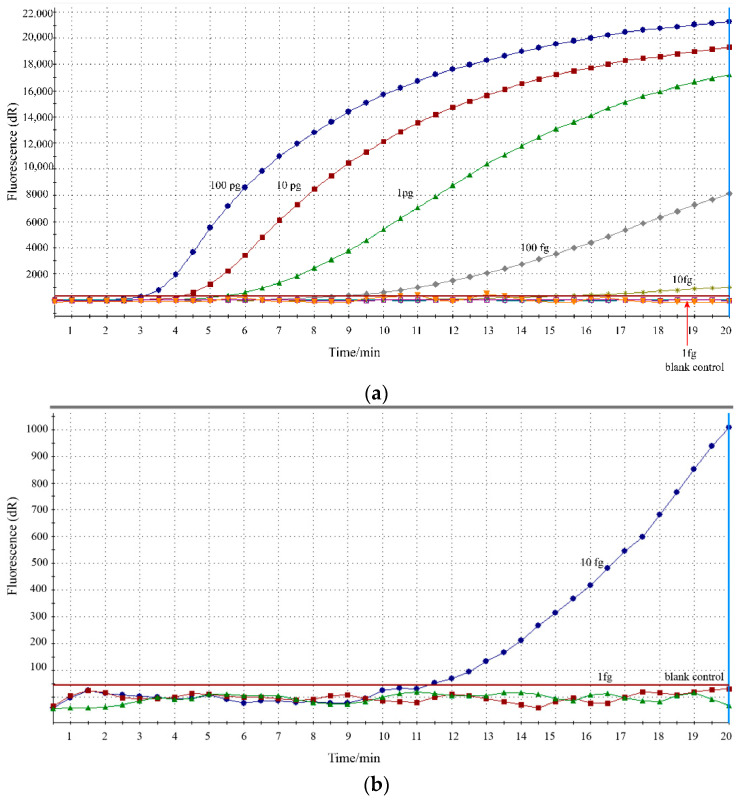
Analytical sensitivity of the RPA assay. (**a**). Sensitivity evaluation was conducted with 10-fold dilutions of M129 DNA templates ranging from 100 pg/μL to 1 fg/μL; (**b**). The LOD of the RPA assay was 10 fg/μL for *M. pneumoniae* detection.

**Table 1 jcm-11-01780-t001:** The sequence of primers and probes for *M. pneumoniae* RPA assay.

Primers/Probes	Sequence (5′-3′)	Amplicon Size (bp)	Gene
Forward primer	TGACACCGCAAGACAGTGCAATAACTCAGT		
Reverse primer	CTGAACATCAACAAAGAAGGTGCTAGCTGC	179 bp	CARDS *
probe	ATACCAAGAGTGGTTCACAACACGATT/i6FAMdT//idsp//Ibhq1dt/ATGTATGTCCTTTG ^#^		

^#^ i6FAMdT-6-carboxyfluorescein labeled dT nucleotides; idsp-tetrahydrofuran; bhq-black hole quencher; ibhq1dt-bhq1 labeled dt nucleotides. * The NCBI Reference Sequence of CARDS gene is NZ_CP010546.1.

**Table 2 jcm-11-01780-t002:** The information of strains used in this study.

Strains	Source of Strains	Number of Strains	RPA Results
*Mycoplasma pneumoniae*	M129	1	P
*Mycoplasma genitalium*	ATCC33530	1	N
*Mycoplasma orale*	ATCC23714	1	N
*Mycoplasma hominis*	ATCC23114	1	N
*Mycoplasma penetrans*	ATCC55252	1	N
*Mycoplasma primatum*	ATCC25960	1	N
*Streptococcus pneumoniae*	Isolated strain (BCH)	1	N
*Staphylococcus aureus*	Isolated strain (BCH)	1	N
*Klebsiella pneumoniae*	Isolated strain (BCH)		N
*Pseudomonas aeruginosa*	Isolated strain (BCH)	1	N
*Mycobacterium tuberculosis*	Isolated strain (BCH)	1	N
*Haemophilus influenzae*	Isolated strain (BCH)	1	N
*Acinetobacter baumannii*	Isolated strain (BCH)		N
*Stenotrophomonas maltophilia*	Isolated strain (BCH)	1	N
*Bordetella pertussis*	Isolated strain (BCH)	1	N
*Legionella pneumophila*	Isolated strain (BCH)	1	N
*Respiratory syncytial virus*	Isolated strain (BCH)	1	N
*Adenovirus type 3*	Isolated strain (BCH)	1	N
*Rhinovirus*	Isolated strain (BCH)	1	N
*H1N1 influenza*	Isolated strain (BCH)	1	N

M129, *M. pneumoniae* reference strains; P, positive; N, negative. BCH, Beijing Children’s Hospital; ATCC, American Type Culture Collection.

**Table 3 jcm-11-01780-t003:** Comparison of RPA assay and real-time PCR for *M. pneumoniae* detection.

RPA	Real-Time PCR		Total	Kappa	Performance of RPA Assay in Comparison to Real-Time PCR	
	Positive	Negative			Sensitivity	Specificity
Positive	88	4	92	0.958	100%	96.1%
Negative	0	98	98			
Total	88	102	190			

## Data Availability

Not applicable.

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
