# Peer review of "Recombinase Polymerase Amplification Combined with Real-Time Fluorescent Probe for Mycoplasma pneumoniae Detection"

_jcm, 2022, doi:10.3390/jcm11071780_

Round 1
Reviewer 1 Report
Recombinase polymerase amplification combined with real-2 time fluorescent probe for Mycoplasma pneumoniae detection. Journal of clinical medicine
Mycoplasma pneumoniae is a wall-less bacteria responsible for atypical pneumonia in children. This is a world-wide pathogen and its detection is important because (1) it could lead to severe pneumonia and extra-pulmonary complications and (2) it is naturally resistant to beta-lactamin family. Thus if M. pneumoniae is detected, the antibiotherapy have to be adapted.
The article proposed by Ting-ting Jiang et al. aims to set up a fast, sensible and specific method for the detection of M. pneumoniae. The authors developed a recombinase polymerase amplification (RPA) of the card gene (which codes for the M. pneumoniae toxin) coupled with a fluorescent probe for the detection of the amplification product.
The paper is well written, the objectives are clearly defined and the results are well presented. It will be interesting to test the sensitivity of the RPA with negative sputum artificially contaminated with several concentrations of M. pneumoniae (see point 11).
I have only few questions and remarks.
Materials and methods:
1-Table 1 and Table 3 are not cited in the text. Line 86: lists of mycoplasmas, virus and bacteria are linked to Table 2 instead of Table 1.
Please checked throughout the text the order of appearance of the tables and their corresponding number.
2-line 77: please define "FAM"
3-line 82: please name the fluorescence detector (manufacturer, model,…). I think this is a worthy information for readers, as this detector performs the incubation and the fluorescence quantification. What is the temperature precision for this apparatus?
4-For the RPA assay: Please precise that this is an end point reading for the interpretation of the result. What is the red (figure 1) or blue (figure 2) line at the bottom of the graphics? If it is a limit of fluorescence detection, how it is determined?
5-line 86: It is not "5 other Mycoplasmas strains" but "5 other Mycoplasma species"
6-lines 90-91: the limit of detection was determined as the last dilution giving a positive result. In the experiment, it was 10 fg, which is very close to the red line after 20 min. How a positive result is determined?
7-line 95, what is the name and references of the commercial real-time PCR ? Is this test used in hospital for M. pneumoniae detection? What is the target of the RT PCR ?
Results:
8-Please check the figures order and their corresponding numbers, it seems that figure 1 is figure 3? Legends are corrects.
9-The graph regarding temperature is a little beat small.
10-Lines 125-129. I think that the comparison between RPA and the RT-PCR could be completed by several data: number of samples RPA+/PCR+, number of samples RPA+/PCR-, number of samples RPA-/PCR+ and number of samples RPA-/PCR-. This could give a better idea of the correlation between the two tests.
Discussion:
11- line 139: please remove "excellent" and "outstanding". Specificity and sensibility for the RPA were determined and the results showed that RPA is sensible and specific. However, please note that:
-(1) sensibility and specificity of the RPA were compared with those of the RT PCR, LAMP and MCDA only from data retrieved from corresponding papers i.e. different samples, experimental conditions and apparatus. If you want to compare sensibility and specificity between two tests, it is better to do the two tests on the same samples and the same experimental conditions.
-(2) artificial contamination of negative sputum is a better way for determining the RPA sensibility.
Author Response
Responses to the reviewers
Reviewer 1:
- Table 1 and Table 3 are not cited in the text. Line 86: lists of mycoplasmas, virus and bacteria are linked to Table 2 instead of Table 1.
Please checked throughout the text the order of appearance of the tables and their corresponding number.
Response: Thank you for your careful check. All the tables and figures were rearranged and linked correctly in the manuscript.
- line 77: please define "FAM"
Response: The definition of “FAM” has been added. Please see Page 4 line 86-87. “FAM (Carboxyfluorescein)”.
- line 82: please name the fluorescence detector (manufacturer, model,…). I think this is a worthy information for readers, as this detector performs the incubation and the fluorescence quantification. What is the temperature precision for this apparatus?
Response: The name of the fluorescence detector has been added. Please see Page 4 line 92. “fluorescence detector (Agilent Technology, AriaMx Real-time PCR)”. The temperature precision for this apparatus is ±0.2℃ according to manufacturer’s instruction.
- For the RPA assay: Please precise that this is an end point reading for the interpretation of the result. What is the red (figure 1) or blue (figure 2) line at the bottom of the graphics? If it is a limit of fluorescence detection, how it is determined?
Response: The description of fluorescence acquisition has been added. Please see Page 4 line 86-88. “Fluorescence acquisition of FAM (Carboxyfluorescein) fluorophore was performed every 30 s (end point reading) for 20 minutes at the temperature of 39℃.”
In Figure 1 and Figure 2, the bottom straight lines represent the threshold of fluorescence detection. It is determined by just exceeding the highest point of amplification curve of negative control.
- line 86: It is not "5 other Mycoplasmas strains" but "5 other Mycoplasmaspecies"
Response: The description has been revised as recommended. Please see Page 4 line 97.
- lines 90-91: the limit of detection was determined as the last dilution giving a positive result. In the experiment, it was 10 fg, which is very close to the red line after 20 min. How a positive result is determined?
Response: A positive result was determined as follows: The reaction can be completed within 20 min and the amplification curve was above the threshold line (described above) and S-shaped. In Figure 3, the amplification curve of 10 fg will be obvious (its fluorescence is 1000) after the removal of other curves of high concentrations. Please see modified Figure 3b.
- line 95, what is the name and references of the commercial real-time PCR ? Is this test used in hospital for M. pneumoniaedetection? What is the target of the RT PCR ?
Response: The name of the commercial real-time PCR is M. pneumoniae and macrolide resistance detection kit. The manufacturer was described in Page 5 line 109. “Mole Bioscience, Jiangsu, China”. This kit is used in clinical work in both Beijing Children’s Hospital and Baoding Children’s Hospital. The target of real-time PCR is P1 gene of M. pneumoniae.
In the published studies, this commercial kit was often used as a reference.
1) Increased Macrolide Resistance Rate of M3562 Mycoplasma pneumoniae Correlated With Macrolide Usage and Genotype Shifting. Front Cell Infect Microbiol. 2021, 11: 675466. doi: 10.3389/fcimb.2021.675466.
2) The clinical characteristics of macrolide-resistance Mycoplasma pneumonia pneumonia in children: a case-control study. Chin J Evid Based Pediatr. 2016, 11(5): 357-360.
- Please check the figures order and their corresponding numbers, it seems that figure 1 is figure 3? Legends are correct.
Response: The order of figures and their corresponding numbers are checked and revised.
- The graph regarding temperature is a little beat small.
Response: The graph regarding temperature screening has been enlarged for better illustrating. Please see Figure 1.
- Lines 125-129. I think that the comparison between RPA and the RT-PCR could be completed by several data: number of samples RPA+/PCR+, number of samples RPA+/PCR-, number of samples RPA-/PCR+ and number of samples RPA-/PCR-. This could give a better idea of the correlation between the two tests.
Response: The data was presented as recommended. Please Page 6 line147-149, Page14 line 324-326.
- line 139: please remove "excellent" and "outstanding". Specificity and sensibility for the RPA were determined and the results showed that RPA is sensible and specific. However, please note that:
Response: The sentence was revised as recommended. Please see Page 6 line 154-155. “The established RPA assay in this study is sensitive and specific which can be completed within 20 minutes……”
- sensibility and specificity of the RPA were compared with those of the RT PCR, LAMP and MCDA only from data retrieved from corresponding papers i.e. different samples, experimental conditions and apparatus. If you want to compare sensibility and specificity between two tests, it is better to do the two tests on the same samples and the same experimental conditions.
Response: I agree with your views. It is the best way to compare the sensitivity and specificity of different tests using the same samples. However, we only compared the LOD of different methods in this study. The limitation was added in the section of limitations. Please see Page 6 line 165-168.
“However, the better way to compare the sensitivity and specificity between two tests is using the same samples and the same experimental conditions. Further study is needed to evaluate the clinical performance of these different methods.”
- artificial contamination of negative sputum is a better way for determining the RPA sensibility.
Response: I agree with your opinion. However, the M. pneumoniae strains with exact copy numbers will be needed to spike into the sputum. It is a little difficult for us to culture M. pneumoniae and count the CCU. Thus, we only extracted the DNA of M. pneumoniae and made the dilutions to determine the LOD of RPA assay in this study.
Reviewer 2 Report
Jiang et al. hereby presented another intriguing method for detecting M.pneumoniae from clinical specimens.
However, as a manuscript, the Reviewer found some points to be pointed out.
For detection methods of the pathogen, mentioned in the Introduction section, lateral flow immunochromatography based methods, used in some countries, accompished within 15minutes, are not mentioned at all.
In the Materials and Methods section, a number of statements are incomplete; for example, gene sequences lack proper reference to accession numbers, the RPA reaction volume sems inconsistent between template conditions, manufacturers of apparati sometimes absent, the starting material as well as the quality of original M129 genome DNA extract solution uncertain, the diluent for serial dilution also uncertain, etc. The table with oligonucleotide sequence also lacks appropriate explanation for the audience to understand the table easiliy.
In the Results section, the figures mentioned does not necessarily seem to represent what is described in the text. Some sentences in the Results section mention what to be presented in the Discussuions section. Concerning statistics, there is only one occasion where statistics is mentioned, whereas two methods are mentioned in the M&M; mentioning only the method(s) that appear in the manuscript is sufficient.
In the Discussion section, the authors compare the LOD of their method with commercial real-time PCR. However, they did not carry out direct comparison using identical analytes side-by-side. This comparison is not scientifically convincing, since carrying out comparison using the same analyte is not so difficult to carry out. The LOD stated as 10fg is also still ambiguous, unless the actual reactions of all pentuplated reactions are displayed.
Sincere prudent scientific discussion might be encouraged.
Author Response
Reviewer 2:
- For detection methods of the pathogen, mentioned in the Introduction section, lateral flow immunochromatography based methods, used in some countries, accompished within 15minutes, are not mentioned at all.
Response: Thank you for your suggestion. We agree that lateral flow immunochromatography based methods perform well for M. pneumoniae detection. However, these methods belong to serological testing, possessing the same shortcomings as routine serology test. Thus, we didn’t mention that in the introduction.
- In the Materials and Methods section, a number of statements are incomplete; for example, gene sequences lack proper reference to accession numbers, the RPA reaction volume sems inconsistent between template conditions, manufacturers of apparati sometimes absent, the starting material as well as the quality of original M129 genome DNA extract solution uncertain, the diluent for serial dilution also uncertain, etc. The table with oligonucleotide sequence also lacks appropriate explanation for the audience to understand the table easiliy.
Response: Thank you for your suggestions.
(1) The accession number of gene sequences has been added. Please see Page 12 line 317. “The NCBI Reference Sequence of CARDS gene is NZ_CP010546.1.”
(2) The description of RPA reaction system has been revised. Please see Page 4 line 85-86. “…… and added distilled water (DW) to the total volume of 50 μL.”
(3) The manufactures of the apparatus used have been added. Please see Page 4 line 92, “fluorescence detector (Agilent Technology, AriaMx Real-time PCR)” and Page 5 line 100, “Nanodrop ND-1000 (Thermo, America)”.
(4) The quality control of M129 genome DNA was added. Please see Page 5, line 100-101. “The DNA template of reference strain M129 was quantified using Nanodrop ND-1000 (Thermo, America), presented by A260/280 ratio. A ratio ranging from 1.8 to 2.0 was acceptable.”
(5) The diluent for serial dilution has been added. Please see Page 5 line 101-102. “Then the DNA template was serially diluted to 100 pg, 10 pg, 1 pg, 100 fg, 10 fg, and 1 fg per microliter using DW.”
(6) The table 1 with oligonucleotide sequence was modified for easy understanding. Please see Page 12 line 313-317.
- In the Results section, the figures mentioned does not necessarily seem to represent what is described in the text. Some sentences in the Results section mention what to be presented in the Discussuions section. Concerning statistics, there is only one occasion where statistics is mentioned, whereas two methods are mentioned in the M&M; mentioning only the method(s) that appear in the manuscript is sufficient.
Response: We have checked and corrected the orders of the figures. The sentences repeated in the results and discussions section have been modified. Please see Page 6 line 134-137 and Page 8 line 194-199.
The description of statistical methods was revised. Please see Page 5 line 116-121.
“SPSS version 23.0 (IBM) was used for all the statistical analysis. Categorical variables were presented as frequencies. The diagnostic performance of RPA assay was presented as sensitivity and specificity. The sensitivity of the two methods was compared by the Chi-square test. P<0.05 was considered statistically significant. The kappa value of the real-time PCR and RPA assays was calculated. The Kappa value was interpreted as follows: poor, <0.4; moderate, 0.4-0.75; perfect, >0.75.”
- In the Discussion section, the authors compare the LOD of their method with commercial real-time PCR. However, they did not carry out direct comparison using identical analytes side-by-side. This comparison is not scientifically convincing, since carrying out comparison using the same analyte is not so difficult to carry out. The LOD stated as 10fg is also still ambiguous, unless the actual reactions of all pentuplated reactions are displayed.
Response:The LOD of commercial real-time PCR (500 copies/mL, about 45 fg per reaction) listed in the discussion (Page 7 line 162) comes from the manufacturer’s instruction. In addition, we also did the serial dilutions of M129 DNA templates using commercial real-time PCR. The results were shown below.
I agree with your opinion that the LOD of RPA assay was not stated exactly, as we did not do all the concentrations. However, using serial dilutions to evaluate the LOD of a specific method is also a common method in the literature.